# Assessment of Uptake, Accumulation and Degradation of Paracetamol in Spinach (*Spinacia oleracea* L.) under Controlled Laboratory Conditions

**DOI:** 10.3390/plants11131626

**Published:** 2022-06-21

**Authors:** Zarreen Badar, Abdallah Shanableh, Ali El-Keblawy, Kareem A. Mosa, Lucy Semerjian, Abdullah Al Mutery, Muhammad Iftikhar Hussain, Sourjya Bhattacharjee, François Mitterand Tsombou, Sefeera Sadik Ayyaril, Islam M. Ahmady, Attiat Elnaggar, Muath Mousa, Mohammad H. Semreen

**Affiliations:** 1Research Institute of Science and Engineering (RISE), University of Sharjah, Sharjah P.O. Box 27272, United Arab Emirates; akeblawy@sharjah.ac.ae (A.E.-K.); safeeraayyaril@sharjah.ac.ae (S.S.A.); atyatnajar@yahoo.com (A.E.); mmousa2@sharjah.ac.ae (M.M.); 2Department of Civil and Environmental Engineering, University of Sharjah, Sharjah P.O. Box 27272, United Arab Emirates; sbhattacharjee@sharjah.ac.ae; 3Department of Applied Biology, College of Sciences, University of Sharjah, Sharjah P.O. Box 27272, United Arab Emirates; kmosa@sharjah.ac.ae (K.A.M.); aalmutery@sharjah.ac.ae (A.A.M.); tsombou@yahoo.fr (F.M.T.); ialtayeb@sharjah.ac.ae (I.M.A.); 4Department of Biology, Faculty of Science, Al-Arish University, Al-Arish 45511, Egypt; 5Department of Biotechnology, Faculty of Agriculture, Al-Azhar University, Cairo 11751, Egypt; 6Department of Environmental Health Sciences, College of Health Sciences, University of Sharjah, Sharjah P.O. Box 27272, United Arab Emirates; lsemerjian@sharjah.ac.ae; 7Human Genetics and Stem Cells Research Group, Research Institute of Sciences and Engineering, University of Sharjah, Sharjah P.O. Box 27272, United Arab Emirates; 8Molecular Genetics Lab, Biotechnology Lab, Research Institute of Sciences and Engineering, University of Sharjah, Sharjah P.O. Box 27272, United Arab Emirates; 9Department of Plant Biology and Soil Science, Campus Lagoas Marcosende, Universidad de Vigo, 36310 Vigo, Spain; mih786@gmail.com; 10Departmento de Biología Vegetal, Universidad de Málaga, 29016 Málaga, Spain; 11Department of Botany and Microbiology, Faculty of Science, Alexandria University, Alexandria 21568, Egypt; 12Department of Medicinal Chemistry, College of Pharmacy, University of Sharjah, Sharjah P.O. Box 27272, United Arab Emirates; msemreen@sharjah.ac.ae; 13Sharjah Institute for Medical Research, University of Sharjah, Sharjah P.O. Box 27272, United Arab Emirates

**Keywords:** abiotic stress, spinach, paracetamol, degradation, growth parameters, chlorophyll florescence, photosynthetic pigments, elements, microbes

## Abstract

The occurrence and persistence of pharmaceuticals in the food chain, particularly edible crops, can adversely affect human and environmental health. In this study, the impacts of the absorption, translocation, accumulation, and degradation of paracetamol in different organs of the leafy vegetable crop spinach (*Spinacia oleracea*) were assessed under controlled laboratory conditions. Spinach plants were exposed to 50 mg/L, 100 mg/L, and 200 mg/L paracetamol in 20% Hoagland solution at the vegetative phase in a hydroponic system. Exposed plants exhibited pronounced phytotoxic effects during the eight days trial period, with highly significant reductions seen in the plants’ morphological parameters. The increasing paracetamol stress levels adversely affected the plants’ photosynthetic machinery, altering the chlorophyll fluorescence parameters (F_v_/F_m_ and PSII), photosynthetic pigments (Chl a, Chl b and carotenoid contents), and composition of essential nutrients and elements. The LC-MS results indicated that the spinach organs receiving various paracetamol levels on day four exhibited significant uptake and translocation of the drug from roots to aerial parts, while degradation of the drug was observed after eight days. The VITEK^®^ 2 system identified several bacterial strains (e.g., members of *Burkhulderia*, *Sphingomonas*, *Pseudomonas*, *Staphylococcus*, *Stenotrophomonas* and *Kocuria*) isolated from spinach shoots and roots. These microbes have the potential to biodegrade paracetamol and other organic micro-pollutants. Our findings provide novel insights to mitigate the risks associated with pharmaceutical pollution in the environment and explore the bioremediation potential of edible crops and their associated microbial consortium to remove these pollutants effectively.

## 1. Introduction

The United Arab Emirates (UAE) is situated in the Arabian desert, facing harsh climatic conditions and water scarcity. Such a situation poses a severe constraint to agricultural development. Therefore, alternative irrigation water sources are essential for achieving sustainable agriculture. The UAE is highly dependent on desalination for water supply and ranks among the highest per capita water consumers in the world [1]. Therefore, the reuse of treated wastewater for irrigation is a highly attractive alternative to reduce reliance on desalination and save the limited freshwater resources in the UAE. However, significant barriers exist to the widespread use of treated wastewater for irrigation due to the presence of emerging contaminants, especially pharmaceuticals, that may adversely affect soil quality and/or public health. Although these pollutants have been detected in very low concentrations (ng/L and µg/L), they pose a substantial risk to the environment due to their daily entry into terrestrial ecosystems [2].

Emerging contaminants such as pharmaceuticals have raised concerns in recent years because of the potential for chronic toxicity and the development of resistance to microbial pathogens in humans and ecosystems [3]. The presence of pharmaceuticals, their toxic degradation products, and different microscopic contaminants in reclaimed water poses a new challenge for wastewater professionals and the pharmaceutical enterprise, as wastewater treatment plants are not designed to eliminate them effectively [4]. Despite the fact that many drugs are naturally bioactive compounds, their toxicity to flora is not widely known. Thus, it is important to assess pharmaceutical contaminations and their phytotoxicity in plants.

Studies report that various drugs can enter plant organs via the roots and leaves and induce phytotoxicity, depending on the composition and concentration of the compound and plant species [5]. In addition, deadly doses vary between studies; however, the range of plant toxicity of antibiotics is from 4 to more than 10,000 µg [6]. These pollutants can be detoxified, disrupted, and separated once taken up by roots and translocated to leaves [7,8]. Moreover, the physicochemical properties of pharmaceuticals, physiological characteristics of plants, and plant–pharmaceutical interactions (e.g., water solubility, half-life, sorption affinity) all collectively influence the transport and accumulation of pharmaceuticals in plants [6,9]. 

Paracetamol, also named acetaminophen, is one of the most widely consumed analgesic drugs among over-the-counter (OCT) and prescribed medicines [10]. The global paracetamol market is expected to be valued at 126.2 USD million in 2022 due to the COVID-19 pandemic and is forecast to adjust to 121.7 USD million by 2028 [11]. Paracetamol is considered one of the most rapidly growing contaminants on the planet and has previously been reported in concentrations of up to 200 μg/L in wastewater effluents and 28.70 μg/L in surface waters [12], despite having high removal efficiency and a 90% elimination rate in wastewater treatment plants within 15 days [13,14,15]. 

Researchers have described the strong tendency of paracetamol and other pharmaceuticals towards sorption, accumulation, and persistence in plants and soil sediments [6,16,17], and reported a variety of phytotoxic effects in different organs of plants in soil and hydroponic systems, all of which were influenced by the applied doses [18,19]. According to recent findings, leafy vegetables (such as lettuce, spinach, cabbage, and celery) have the greatest ability to receive and accumulate emerging contaminants in their edible tissues [20,21]. Spinach and lettuce are the most common plants that uptake pharmaceuticals and other emerging contaminants from treated wastewater effluents used for irrigation [22]. In cucumber, a 5 mg/L paracetamol dose inhibited Cytochrome P450 [23], whereas beans treated for three weeks with 1–4 g/L paracetamol promoted phytotoxicity by reducing stomatal conductance and β carotene concentration [24]. Moreover, paracetamol tested at lethal doses of 151 mg/L for 168 h in *Brassica juncea* (Indian mustard) resulted in bleaching and dot-like lesions on the adaxial side of the leaf after 72 h, and necrosis was seen after seven days [25]. However, when paracetamol was applied at the concentration of 151–181 mg/L, it did not display any symptoms of toxicity in *Hordeum vulgare* (barley), *Lupinus luteolus* (pale yellow lupine), and *Phragmites autralis* [26]. Similarly, relatively lower toxic effects and only 12–18% inhibition were reported in *Linum usitatissimum* (flax) and *Aromoratia rusticana* (horseradish) [26].

Recent studies conducted at the Sharjah sewage treatment plant have reported the presence of pharmaceutical pollutants in treated wastewater in urban areas with paracetamol; a concentration of 5235 ng/L paracetamol was reported in effluent wastewaters [27,28]. Despite being classified as a pseudo-persistent drug, paracetamol is discharged into the environment on a regular basis, presenting major issues for both human health and the ecosystem [6,29,30]. Moreover, there is scarce information on the effects of the accumulation of and long-term exposure to minute concentrations of this drug on both human and plant growth and development. 

Thus, spinach (*Spinacia oleracea*), a green vegetable, is the subject of our study, which aims to evaluate the absorption, translocation, accumulation, and degradation of paracetamol in various organs in a controlled laboratory setting. The study also assessed paracetamol’s phytotoxicity on several morphological and physiological traits, and identified several bacterial strains in spinach roots and shoots. We hypothesized that the regular use of wastewater with low concentrations of emerging contaminants (e.g., paracetamol) in crop irrigation would concentrate them in the soil and other environmental matrixes, and result in their uptake into, translocation to and accumulation in different organs of the plants. It was also hypothesized that the accumulated paracetamol and its degradation products in the tissues of the plants would affect essential element absorption and biochemical processes, affecting pigment synthesis and photosynthesis efficiency.

## 2. Results

### 2.1. Paracetamol Uptake and Metabolism in Spinach

The results showed that paracetamol treatments, exposure periods, and their interactions significantly affected paracetamol concentrations in spinach shoots and roots (Table 1, Figure 1, Appendix A). Spinach plants treated with 50 mg/L, 100 mg/L, and 200 mg/L paracetamol accumulated 75 µg/g, 136 µg/g, and 1012 µg/g paracetamol, respectively, after four days in shoot tissues. After eight days, the drug concentrations significantly declined to 56 µg/g, 73 µg/g, and 396 µg/g in the 50 mg/L, 100 mg/L, and 200 mg/L treatments, respectively, indicating the biodegradation of paracetamol after day 4. Paracetamol concentrations in plants treated with 200 mg/L paracetamol paracetamol were significantly greater than in plants treated with 50 mg/L and 100 mg/L over the two treatment periods (Table 1, Figure 1a, Appendix A). 

The accumulated levels of paracetamol were very low in the roots compared to in the shoots. The concentrations in plants’ roots treated with 50 mg/L, 100 mg/L, and 200 mg/L paracetamol were 6.5 µg/g, 7.8 µg/g, and 10.3 µg/g, respectively, after four days, with significant reductions observed in the concentrations (to 2.3 µg/g, 2.6 µg/g, and 3.6 µg/g, respectively) after eight days (Table 1, Figure 1b, Appendix A). The results indicate high translocation of paracetamol after its uptake from the roots to the shoots. The translocation factor in plants treated with 50 mg/L, 100 mg/L, and 200 mg/L paracetamol significantly increased from 11.54, 17.46, and 98.74, respectively, after four days to 24.12, 27.63, and 112.15, respectively, after eight days (Table 1, Figure 1c). However, there were insignificant effects of the interaction between paracetamol treatments and exposure period on the translocation factor (*p ≥* 0.05, Table 1, Figure 1c). 

The dried leaves from plants treated with different paracetamol concentrations were collected and assessed after eight days. On day eight, the concentration of paracetamol in dried leaves treated with 100 mg/L and 200 mg/L paracetamol was greater (143 µg/g and 628 µg/g, respectively) than in fresh leaves (73 µg/g and 396 µg/g, respectively). The accumulation of paracetamol increased in dried leaves with an increase in the applied doses of the pharmaceutical drug (Figure 1d, Appendix A).

### 2.2. Influence of Paracetamol on Plant Growth Parameters

The effect of different paracetamol levels was evident on the leaves, shoot and root lengths, number of leaves, and shoot growth tolerance index (%) (Appendix A, Figure 2). Spinach root length, shoot length and number of leaves were significantly reduced in the 200 mg/L paracetamol group compared to the controls. There was no significant difference between the lower two paracetamol concentrations (50 mg/L and 100 mg/L) in all growth traits (Appendix A, Figure 2a–d). The sensitivity of the growth to paracetamol was organ-specific. For example, the reductions at 50 mg/L, 100 mg/L, and 200 mg/L, compared to controls, were 17.4%, 13.0%, and 34.8%, respectively, in leaf number, 31.0%, 41.4%, and 58.6%, respectively, in shoot length, and 20.7%, 62.3%, and 66.0%, respectively, in root length, indicating that leaf number was least affected and root length was most affected. Moreover, shoot tolerance was significantly lower (*p* ≤ 0.05) at 200 mg/L than root tolerance (Appendix A, Figure 2d). In addition, plants exposed to higher paracetamol concentrations showed leaf margin necrosis and increased numbers of withered leaves and browning in the root tissues (Appendix A).

### 2.3. Impact of Paracetamol on Photosynthetic Pigments and Chlorophyll Fluorescence

Besides detecting possible damage in the photosynthetic apparatus of plants, fluorescence measurements allow for qualitative and quantitative analysis based on the absorption and utilization of light energy [31]. Two-way ANOVA showed significant effects of paracetamol and treatment period and their interaction on the maximum quantum efficiency of PSII and F_v_/F_m_ (Table 2). There was no significant difference in F_v_/F_m_ between four and eight days in the non-treated plants (control). The F_v_/F_m_ values decreased with the increase in paracetamol concentration, but the reduction was more pronounced after eight days than four days. The F_v_/F_m_ of plants treated with 50 mg/L, 100 mg/L and 200 mg/L decreased below the control by 0.1%, 2.2% and 2.6%, respectively, after four days and 9.8%, 9.05%, and 10.3%, respectively, after eight days (Figure 3a). A significant reduction in the quantum yield efficiency of photosystem II was also seen, of 7.87%, 7.25%, and 11.84% after eight days in plants treated with 50 mg/L, 100 mg/L, and 200 mg/L paracetamol, respectively, as compared to four days with 0.13%, 1.67% and 4.25% reductions, respectively (Figure 3b).

The two-way ANOVA results showed significant effects of paracetamol treatment period on Chl a, Chl b, and total chlorophyll (Table 2). Interestingly, Chl a, Chl b, and total chlorophyll significantly increased after eight days compared to after four days. Such differences were more obvious at higher paracetamol concentrations (100 mg/L and 200 mg/L). The results indicate that more prolonged exposure at higher paracetamol concentrations enhanced the chlorophyll synthesis. After eight days, Chl a significantly increased by 25.34% with 200 mg/L paracetamol treatment and Chl b content increased by 4.05% and 33.23% with 100 mg/L and 200 mg/L paracetamol treatments, respectively, compared to the control. However, there were insignificant effects on all the pigments for the different paracetamol treatments and the interactions between periods and treatments (*p ≥* 0.05, Table 2, Figure 3). 

### 2.4. Plant Nutrients and Elements

#### 2.4.1. Macronutrients

There were significant effects of different paracetamol treatments on the concentration of nutrients in spinach shoots and roots (Appendix A, Figure 4). In the roots, there were gradual decreases in the concentrations of two macronutrients (Ca, K) with the increase in paracetamol treatments (Figure 4a,b). In the 50 mg/L, 100 mg/L, and 200 mg/L paracetamol treatments, Ca decreased by 33.1%, 39.9% and 49.8%, respectively, and K decreased by 65.1%, 69.1%, and 76.1%, respectively. For Mg, the concentration decreased by 40.2% and 46.3% in the 50 mg/L and 100 mg/L paracetamol treatments, respectively, but only 13% in the 200 mg/L treatment (Figure 4c). However, the lower paracetamol treatments (50 mg/L and 100 mg/L) significantly reduced the concentrations of the two macronutrients. Additionally, the high paracetamol treatment (200 mg/L) significantly increased the concentrations of Ca and K in the shoots (Figure 4a,b) and reduced the concentrations of Mg in the shoots compared to the control. Still, the reduction in the concentrations of Mg caused by the 200 mg/L treatment was less than that in the 50 mg/L and 100 mg/L treatments (Figure 4c).

#### 2.4.2. Micronutrients

The effects of the paracetamol treatments on the two micronutrients (Fe and Mn) in both roots and shoots were significant (Appendix A). Unlike most of the other elements, the concentration of Fe was significantly greater in roots than in shoots. The concentration was considerably greater in the roots of plants treated with 200 mg/L paracetamol. There was no significant difference in Fe concentration in the shoots of plants across the different paracetamol treatments (Figure 4d).

The Mn concentration was significantly reduced in shoots of plants treated with 100 mg/L and 200 mg/L paracetamol than in the control and 50 mg/L. The concentration of Mn was significantly greater in the shoots than in the roots. In the latter, Mn was significantly lower in plants treated with 50 mg/L than in the control and the higher concentrations (Figure 4e).

#### 2.4.3. Sodium

Sodium (Na) decreased in the roots by 4.9%, 5.4%, and 6.9% in the 50 mg/L, 100 mg/L and 200 mg/L paracetamol treatments, respectively. Moreover, Na also decreased in the shoots of the treated plants as compared to the control plants (Figure 4f).

#### 2.4.4. CHNS Analysis

Elemental analysis of spinach shoots using the CHNS analyzer revealed significant effects of paracetamol on N and S percentages in the spinach shoots (Appendix A). There was a considerable increase in the weight percentage of nitrogen in spinach shoots treated with 200 mg/L of paracetamol (3.84%), which was similar to control shoots (3.88%), and lower in those treated with 50 mg/L and 100 mg/L (2.94% and 2.78%, respectively) as shown in Figure 5. C was higher in shoot samples treated with 50 mg/L (42.01%) and 100 mg/L paracetamol (42.09%) compared to the control (41.03%), which was close to plants treated with 200 mg/L (40.74%). The percentages of H and S detected in all paracetamol-treated shoot samples were similar to control shoots (Figure 5). Overall, the C/N ratios in control and 200 mg/L paracetamol treated shoots samples were similar and lower than those in 50 mg/L and 100 mg/L treated plants (Figure 6).

### 2.5. Microbial Analysis

VITEK^®^ 2 microbial identification system provided rapid identification of the bacterial flora in spinach roots and shoots (Table 3 and Table 4). Among the identified bacterial strains, members of *Burkhulderia, Sphingomonas, Pseudomonas, Staphylococcus, Stenotrophomonas, and Kocuria* were prominent in spinach root and shoot cultures.

## 3. Discussion

Results showed that the effects of paracetamol on spinach uptake, accumulation, degradation, and phytotoxicity were concentration- and organ-dependent. There were obvious deleterious effects of increasing paracetamol stress on spinach growth, morphological and physiological parameters. Leaves displayed symptoms of withering, burning, and necrosis on the margins at higher (100–200 mg/L) paracetamol levels. This result is aligned with observations of a significant decrease in the shoot and root elongation of wheat after 21 days of application of 1.4–22.4 mg/L paracetamol treatments [49]. A similar toxic effect of higher concentrations (>10 mg) of paracetamol was also seen in cucumber plants after seven days of exposure, with a significant reduction in the biomass of leaves and roots [50]. Growth reduction is observed to be an adaptive survival mechanism for plants to endure the oxidative stress damage caused to the cellular components [51].

The photosynthesis machinery of spinach was highly susceptible to paracetamol and was impaired due to the xenobiotic stress. A significant decline occurred in the chlorophyll fluorescence parameters with time. Our results are consistent with a previous finding [24], demonstrating that PSII, PSI, and electron transport in the bean plants were inhibited by paracetamol stress, causing a potential decrease in the photosynthetic activity. A similar decreasing trend was observed in chlorophyll fluorescence for the model macrophyte *Lemna minor* used for environmental risk assessment; when treated with paracetamol, its photosynthetic activity significantly decreased by up to 37% in comparison with the control [16]. Moreover, a 40% reduction in the quantum yield of photosystem II was also reported in maize treated with 10 mg/L paracetamol [52].

Our study indicates an inconsistency in the response of chlorophyll content and photosynthetic efficiency of spinach plants treated with paracetamol. Although paracetamol significantly affected photosynthetic pigments at higher concentrations (>100 mg/L) with the exposure period, it overall decreased the photosynthesis efficiency (Figure 3, Table 2). Similarly, in the macrophyte *Lemna minor*, paracetamol exposure resulted in chlorophyll levels similar to those found in control plants. The endpoint of Chl b was more sensitive to paracetamol than that of Chl a and raised the Chl b content of *L. minor*. However, the F_v_/F_m_ (maximal quantum yield of PSII) was not significantly affected [53]. A recent study [54] also reported that increasing paracetamol concentration and exposure time resulted in non-significant increases in total chlorophyll contents, photosynthetic capacity and photosynthetic rates, but significantly increased the photochemical reflectance index in lettuce. In addition, it has been reported that a lower concentration of Panadol (1.0 mg/L), another paracetamol, enhanced chlorophyll content and photosynthetic efficiency of *Vigna radiate* plants [55]. The authors indicated the possibility of using a lower concentration of Panadol as a plant growth regulator. Similarly, a significant increase in the chlorophyll (27.46%) and carotenoids (41.8%) was found with 500 μM chronic treatment with paracetamol in lettuce. With increasing concentrations of paracetamol, the chlorophyll content also increased. However, high concentrations of paracetamol had an inhibitory effect on the photosynthetic activity of plants and reduced the rate of photosynthesis in the experimental variants compared to the control plants [56].

In nature, plants employ various stress-tolerance strategies to combat biotic and abiotic stresses in their surrounding environment by altering gene expression, protein synthesis, and post-translational modifications that aid in the re-establishment of cellular homeostasis for their survival [51]. Our results showed a significant reduction in the growth tolerance index of spinach shoots. However, the photosynthetic pigments in paracetamol-treated plants increased with the increase in paracetamol concentration and exposure period. Interestingly, the leaves became noticeably greener at the end of the exposure time relative to the control plants. It has been reported that chlorophyll biosynthesis might offset the reduction in electron transport efficiency and Calvin cycle activities under prolonged paracetamol stress [56,57]. The chlorophyll content of cells can act as a protective mechanism against induced oxidative stress, scavenging the accumulated ROS [58]. In addition, the significant increase in the carotenoid content could help deactivate excited chlorophyll and safeguard the photosynthetic system through modulation of lipid peroxidation products, thereby attenuating and/or preventing ROS-induced damage to the photosynthetic system [59]. Still, the increases in levels of chlorophyll and carotenoids were not enough to reduce the ROS-induced damage in spinach plants, especially shoots, treated with 200 mg/L for eight days [60,61]. At 200 mg/L, spinach plants showed lower shoot tolerance with the appearance of several morphological anomalies, such as leaf withering, burning, and necrosis (Appendix A).

Organic xenobiotics must pass via the stomata or traverse the epidermis, which is covered by the cuticle, to permeate into a leaf. Stomata are found on the leaf’s lower (abaxial) side, whereas the thicker cuticle layer is located on the upper (adaxial) side. The stomatal system regulates the penetration of organics into the leaves by having multiple openings that can be enlarged as needed. The plant regulates the intake of chemicals of various molecular masses through the movement of two guard cells that control the opening and closing of the stomata by adjusting the diameter of the opening. K plays a crucial role in regulating the movement of these guard cells, cell elongation, and other vital physiological activities. A rise in K^+^ concentration opens the stoma, facilitating the movement of the xenobiotic into the leaves [62]. Spinach is categorized as a nitrate accumulator, and vegetables grown on sewage sludge-contaminated soil may accumulate nitrate in edible plant portions due to extremely high nitrogen levels in the soil created by the sludge [63]. Additionally, increased potassium uptake is also reported to facilitate the uptake and transport of nitrate to the plant’s aerial parts, improve nitrate metabolism and utilization, and have an indirect effect on chlorophyll synthesis [64,65].

Our results indicated significant increases in the major essential nutrients and elements, such as Ca, K, Fe, and N, in spinach plants treated with higher concentrations (100-200 mg/L) of paracetamol compared to the controls. These essential ions and elements regulate the cellular processes, are involved in chlorophyll production, and possibly contribute to photosynthesis, respiration, oxygen transport, and gene regulation, affecting plant growth and development [66]. In addition, increases in micronutrients, such as Fe and Zn, significantly increased antioxidant activities and reduced oxidative stress, enhancing chlorophyll content and gaseous exchange attributes in spinach plants irrigated with tannery wastewater with chromium [60]. A progressive decline was observed in the Na levels of both shoot and root tissues of plants treated with increasing levels of paracetamol (Figure 4f), likely supporting plants’ adaptation to abiotic stress and affecting the absorption and/or distribution of organic solutes and vital nutrients in the plants [67].

Plant detoxification of xenobiotics is similar to mammalian detoxification; following a phase I activation event, a phase II conjugation process with hydrophilic molecules such as glutathione or glucose occurs. The phase III reactions of xenobiotic detoxification include xenobiotic conjugate storage, degradation, and transfer [9,64]. Conjugation with sugar is one of the main xenobiotic detoxification processes in plants. Huber et al. (2009) pioneered the discovery of paracetamol conjugates in plants, involving a dual detoxification mechanism. They discovered two metabolic pathways in the hairy root culture of horseradish, which culminates in the synthesis of glutathione and a glucose conjugate [68]. However, the fate of drugs in spinach is yet unknown.

Our results showed that the highest accumulations of paracetamol were observed after four days in the shoots and roots of spinach plants treated with 200 mg/L (Figure 1a,b and Appendix A), and in dry leaves assessed after eight days (Figure 1d, Appendix A). Additionally, after eight days, significant reductions of 59.06% and 60.93% in paracetamol concentrations were observed in spinach shoots after 100 mg/L and 200 mg/L paracetamol treatments, respectively (Figure 1a, Appendix A). A similar trend of degradation with time was observed in the roots, but paracetamol accumulation was low (Figure 1b, Appendix A). The higher translocation factor of paracetamol in spinach shoots (Figure 1c) indicates its high mobility into the aerial parts of spinach. In addition, the significant reduction in paracetamol levels after eight days indicates its conjugation ability. 

Our findings coincide with a previous investigation carried out on *Typha latifolia* plants, reporting the uptake and accumulation of paracetamol to be significantly higher (0.077 μg/g FW) in leaf tissues at the end of the first week of 1 mM Paracetamol treatment, indicating a 78% transfer of the total amount. Then, after 30 days, a gradual decrease in the concentration was seen, and only 30% of the drug was present in the shoots [69]. Other studies reported a reduction in the paracetamol concentration immediately after its uptake in cucumber [50], *Solanum nigrum* [70], and *Lemna minor* [16]. The cucumber plants took up paracetamol and conjugated it quickly with glutathione. The paracetamol–glutathione conjugates in cucumber plants exposed to 5 mg/L for 144 h were 15.2 nmol/g and 1.2 nmol/g in cucumber roots and leaves, respectively [50].

Endophytes are nonpathogenic bacteria or fungi that naturally inhabit almost all plant species. Endophytes have a great ability to break down xenobiotics in plants, allowing them to withstand stress under unsuitable soil conditions [71,72]. The inoculation of endophytic bacteria enhances the phytoremediation ability of several plants for remediation of polluted soil and water [73,74]. For example, the inoculation of *Leptochloa fusca* with endophytic bacteria enhanced the biodegradation of organic and inorganic pollutants, decreasing pollutants’ toxicity [74] and promoting plant growth. Although the microbial removal of paracetamol seems to be an effective remediation technique, few bacterial strains, including *Pseudomonas*, *Burkholderia*, *Stenotrophomonas*, *Pseudomonas*, *Rhodococcus***,**
*Bacillus*, *Delftia*, *Kocuria*, *Staphylococcus*, *Acinetobacter* and *Sphingomonas*, are capable of degrading paracetamol and other organic pollutants [35,75,76,77,78]. These microorganisms can use paracetamol as their sole carbon, nitrogen, and energy source.

Moreover, inoculated endophytic bacteria can also regulate the metabolic activities of organic pollutants and horizontal gene transfer from native endophytes [79]. The metabolic pathways of biodegradation, however, remain poorly characterized. Two key metabolites identified during microbial degradation of paracetamol, hydrolytic phenolic dead-end metabolite hydroquinone and 4- aminophenol, were described as carcinogenic and highly toxic compounds. Furthermore, paracetamol phenolic derivatives can cause DNA cleavage and mutagenesis in animal cell lines [76,78].

VITEK 2 provided quick and accurate identification of the microbes contributing to paracetamol degradation. We also identified bacterial strains in paracetamol-treated spinach shoots and roots capable of removing paracetamol (Table 3 and Table 4). The isolated bacteria belong to the genera *Pseudomonas, Sphingomonas, Burkholderia, Staphylococcus, Stenotrophomonas,* and *Kocuria* [35,76,78]. It has been reported that a consortium of *Stenotrophomonas* and *Pseudomonas* microbial strains could degrade paracetamol up to 4 g/L. In contrast, pure cultures of the strains degraded paracetamol completely at 0.4 g/L, 2.5 g/L, and 2 g/L, respectively [77]. The consortium also had significantly higher degradation rates and greatly improved tolerance to paracetamol with a shorter lag time. This also raises serious concerns regarding the development of drug-resistant strains and promotion of the growth of opportunistic pathogens [43,44]. Hence, apart from the beneficial role of bacteria in bioremediation, their antimicrobial resistance and pathogenic potential cannot be underestimated.

## 4. Material and Methods

Pure analytical grade paracetamol (>98% purity) was purchased from Sigma-Aldrich. All solvents and chemical reagents used were of analytical grades. 

### 4.1. Seed Germination and Seedling Development

Spinach (*Spinacia oleracea* L. cv. Matador) seeds were purchased from a local agricultural shop. Seeds were soaked in water for one hour to increase germination potential and then surface sterilized using 5% commercial bleach for 5–7 min, rinsed thrice with distilled water, air-dried, and soaked in pots (9 cm diameter and 7 cm height) containing approx. 350 g of garden organic potting soil. The plants were grown in a CONVIRON plant growth chamber (model E-15) adjusted to a 25/15 °C day/night regime. The lighting in the chamber was white light (1400 µmol/m^2^/s of 167 photosynthetically active radiation) provided by five (400 W) metal halide and five (400 W) high-pressure sodium lamps. 

### 4.2. Hydroponic Cultivation and Paracetamol Exposure

A hydroponic system setup, as described previously [79] was used for the experiments. Briefly, four seedlings with two true leaves were removed from the soil pots, thoroughly rinsed with distilled water, and transferred into glass jars containing 20% Hoagland nutrient solution (Hoagland’s No. 2 basal salt mixture). The jars were covered with aluminum foil to prevent roots from exposure to light and then kept in the growth chambers. The culture jars were aerated frequently, and the nutrient solution was replenished as needed to maintain nutrient solution balance. All experimental material and distilled water used to prepare treatment solutions was autoclaved at 121 °C for 30 min before use to minimize the risk of contamination.

After two weeks of adaptation and upon reaching the 4-6 true leaf stage, plants were exposed to three levels of paracetamol (50 mg/L, 100 mg/L, and 200 mg/L) in 20% Hoagland nutrient solution for an experimental period of 8 days. All treatments were performed in triplicate; each replicate was a mix of four plants in a jar. The treatment solutions were replaced every four days to maintain the same stress levels. Additionally, +ve controls (20% Hoagland nutrient solution in jars with plants) and –ve controls (20% Hoagland nutrient solution without plants) were also included in the experiments in triplicates to determine the abiotic losses of the pharmaceutical. The jars were organized in a randomized block design, with treatment as the main block. In addition, the jars of the different blocks were randomized every two days. Spinach plants were harvested at two time points, i.e., after four days and eight days of paracetamol treatments, to determine the fate of the xenobiotic. Plants were thoroughly washed with distilled water, dried, separated into shoots and roots, weighed, and stored at −80 °C until further analysis.

### 4.3. Assessment of Plant Growth Parameters

Spinach shoot and root lengths were measured using a standard scale, and the number of leaves was counted manually. Fresh weight (FW) of plants (shoots and roots separated) was assessed at harvesting, and dry weight (DW) was calculated after freeze-drying the samples at −84 °C in Freeze Dryer (Labconco Freezone- 6L).

The plant samples were lyophilized, and the growth tolerance indexes for shoots (GTIS %) and roots (GTIR %) were determined based on their respective dry weights [80], as follows:(1)$$\mathrm{GTIS}\%=\frac{\mathrm{Average}\;\mathrm{dry}\;\mathrm{weight}\;\mathrm{of}\;\mathrm{shoots}\;\mathrm{grown}\;\mathrm{in}\;\mathrm{media}\;\mathrm{containing}\;\mathrm{paracetamol}\;}{\mathrm{Average}\;\mathrm{dry}\;\mathrm{weight}\;\mathrm{of}\;\mathrm{shoots}\;\mathrm{grown}\;\mathrm{on}\;\mathrm{media}\;\mathrm{without}\;\mathrm{paracetamol}}*100$$
(2)$$\mathrm{GTIR}\%=\frac{\mathrm{Average}\;\mathrm{dry}\;\mathrm{weight}\;\mathrm{of}\;\mathrm{roots}\;\mathrm{grown}\;\mathrm{on}\;\mathrm{media}\;\mathrm{containing}\;\mathrm{paracetamol}\;}{\mathrm{Average}\;\mathrm{dry}\;\mathrm{weight}\;\mathrm{of}\;\mathrm{roots}\;\mathrm{grown}\;\mathrm{on}\;\mathrm{medium}\;\mathrm{without}\;\mathrm{paracetamol}}*100.\;$$

Chlorophyll Fluorescence Analysis: The initial fluorescence (F_o_), variable fluorescence (F_v_), maximum fluorescence (F_m_), potential quantum efficiency of photosystem II (ΦII), (F_v_/F_m_), and (F_v_/F_o_) were recorded from the third middle section of fully expanded leaves exposed to light, using a pulse-modulated fluorescence monitoring system (FMS-2, Hansatech Instruments Ltd., Norfolk, UK). Measuring setup and calculation of selected parameters were adopted from Hussain and Reigosa (2021) [81]. After the measurements, the same leaves were collected to analyze photosynthetic pigments.

Quantification of the Photosynthetic Pigments: Three replicates of 50 mg leaves, each from four plants from a jar, were immersed in 15 mL falcon tubes containing 5 mL of methanol. The tubes were wrapped with aluminum foil and kept at 4 °C for 48 h until complete bleaching of the leaf occurred to extract the photosynthetic pigments. After complete extraction, 200 μL of supernatant was pipetted in triplicate into 96 well plates and blank wells with methanol. The absorbance of the extracts was measured using an absorbance microplate reader (E.L. x 808- Biotek) at wavelengths of 666 for chlorophyll a, 653 for chlorophyll b, and 470 for carotenoid (Cx+c). According to Lichtenthaler (1987), the concentrations of the pigments were calculated using the following absorption coefficients [82], with V being the extract volume (mL) and W the leaves fresh weight (g):Chlorophyll a (mg/g) = [15.65 × (A666) − 7.34 × (A653)] × V/(1000 × W)
Chlorophyll b (mg/g) = [27.05 × (A653) − 11.21 × (A666)] × V/(1000 × W)
Total Chlorophyll (mg/g) = Chlorophyll a + Chlorophyll b
Cx + c (mg/g) = [1000 × (A470) − 2.86 × (Chla) − 129.2 × (Chlb)]/245 × V/(1000 × W)

### 4.4. The Fate of Paracetamol in Spinach

We followed a previously reported [83] extraction method for determination of paracetamol concentration from plant shoot and root tissues. Briefly, 100 mg of lyophilized and ground spinach shoots and roots samples were extracted in 4 ml EDTA (150 mg/L) and vortexed for 1 min. Then, 5 mL of a mixture of ACN: MeOH (65:35) was added and vortexed for 2 min following the addition of 3 gm Na2SO4 and 0.5 gm NaCl with a 1.5 min vortex. The mixture was kept overnight at 4 °C, then centrifuged the next day at 10,000× *g* for 20 min. The supernatants were filtered through 0.22 µm sterile PES membrane syringe-driven filters (Jet-Biofil, Guangzhou, China) for LC/MS analysis. 

#### 4.4.1. Liquid Chromatography–Mass Spectrometry (LC–MS) Analysis

LC–MS analysis was performed using Waters Acquity UPLC H-Class, Xevo TQD system (Waters Corporation, Milford, M.A, USA) equipped with electrospray ionization operated in the positive ionization mode. The sample injection volume was 10 µL, and chromatographic separation of analytes was carried out on an Acquity UPLC BEH C18 1.7 µm (2.1 × 100 mm) column. Mobile phases consisted of 0.1% formic acid in water (solvent A) and acetonitrile with 0.1% formic acid (solvent B). The run time was 8 minutes, with a retention time of 3.2 min for paracetamol (Appendix A). The gradient separation method is shown in Appendix A.

Calibration standards in the range of 0.5–200 µg/L were prepared with paracetamol in methanol in triplicate, and used to determine the paracetamol concentration in all samples from their respective peak areas. The limit of detection (LOD) and limit of quantification (LOQ) were calculated using the standard deviation (SD) of the responses and the slope (S) of the calibration curve, via the formulas LOD = 3.3 × (SD/S) and LOQ = 10 × (SD/S), yielding the values of LOD = 2.220791163 and LOQ = 6.729670192. The SD of the response was determined based on the standard deviation of y-intercepts of the calibration curve, where LOD= 0.819811731 and LOQ= 2.484277972.

#### 4.4.2. Translocation Factor

The translocation of paracetamol from roots to shoots in different treatments was determined by calculating the translocation factor (TF) [80].
$$\mathrm{TF}=\frac{\mathrm{Paracetamol}\;\mathrm{concentration}\;\mathrm{in}\;\mathrm{shoots}\;}{\mathrm{Paracetamol}\;\mathrm{concentration}\;\mathrm{in}\;\mathrm{roots}}$$

### 4.5. Plant Elemental Analysis

A closed-vessel microwave-assisted digestion procedure assessed the essential nutrients (macro- and micronutrients), and elements in spinach organs [84]. Briefly, 100 mg of lyophilized samples was digested with 5 mL concentrated nitric acid (70% Sigma-Aldrich, Merck, Darmstadt, Germany) and 1 mL H_2_O_2_ (≥30%, Sigma- Aldrich, Merck, Darmstadt, Germany) in PFA Teflon^®^ vessels. The volume was made up to 10 mL with distilled water. The mixture was kept at room temperature for 15–20 min for cold digestion. The vessels were sealed, and samples were heated using an Anton Paar microwave digestion system using the following time and temperature program: ramping to 200 °C for 20 min, and maintaining the temperature at 200 °C for 20 min.

After cooling down to room temperature, the digested solutions were filtered through 0.22 µm sterile PES membrane syringe-driven filters (Jet-Biofil, Guangzhou, China); their volume was made up to 30 mL using Milli-Q water and they were refrigerated at 4 °C. Elemental analysis was carried out via Inductively Coupled Plasma-Optical Emission Spectroscopy (ICP-OES; ICAP 7000, Thermo Fisher Scientific, Cambridge, UK). Paracetamol with 20% Hoagland growth media solution samples (having different paracetamol concentrations) were filtered through 0.22 µm sterile PES membrane syringe-driven filters (Jetbiofil) and diluted with distilled water before ICP-OES analysis.

For simultaneous analysis of carbon (C), hydrogen (H), nitrogen (N), and sulfur (S); a known amount of plant material was oven-dried at 70 °C, finely ground and packed in tin foil capsules and analyzed by an automated Vario MACRO cube CHNS analyzer (PerkinElmer, Waltham, USA). Plant total C, N, H, and S was expressed as the percentage of elements in dried plant material.

### 4.6. Microbial Analysis

Our study employed the VITEK^®^ 2 (BioMerieux, Marcy L’Etoile, France) microbial identification system to identify paracetamol-degrading bacteria in spinach roots and shoots. 

Spinach shoot and root samples were weighed under aseptic conditions under a Class II Biosafety Cabinet, surface sterilized with 70% ethanol for 7 min, and washed three times with sterile distilled water. Then, 2.5% bleach was added to the samples for 10 min and they were washed thrice with sterile distilled water. After sterilization, samples were crushed in 10 mL of sterile distilled water. One milliliter of the extract was taken from each sample and serially diluted to 1 × 10^−5^. Then, from each diluted sample extract, 100 µL was spread on Luria Bertani (LB) agar plates, wrapped with parafilm, and incubated at 28 °C, and bacterial growth was observed for 72 h. Gram’s staining was used for preliminary identification of the isolated bacterial strains. Bacterial suspensions were prepared in sterile saline, and the density was adjusted using VITEK 2 DensiCheck (BioMerieux, Marcy L’Etoile, France) to a McFarland standard of 0.5–0.63. Gram-positive and Gram-negative bacteria were identified using G.P. and G.N. cards, respectively, via VITEK 2 G.N. (21341 BioMerieux, Marcy L’Etoile, France) and G.P. (21342 BioMerieux, Marcy L’Etoile, France) Identification Kits.

### 4.7. Statistical Analysis

Two-way ANOVAs were used to assess the impacts of paracetamol treatments (0 mg/L, 50 mg/L, 100 mg/L, and 200 mg/L) and exposure time (four and eight days) and their interaction on pigments, chlorophyll fluorescence traits and paracetamol concentrations in spinach organs. In addition, ANOVAs were used to assess the impacts of paracetamol concentrations on evaluated parameters, including shoot length, root length, leaf number, growth tolerance index of shoots and roots, plant nutrients and elements. Pairwise comparisons of the means were performed using the post-hoc Tukey’s Honest Significant Differences (HSD) test to identify which pairs of treatments were significantly different. Three biological replicates (each replicate was a mix of four plants in a jar) were used for the assessments. Data were statistically analyzed using SYSTAT (version 13). 

## 5. Conclusions

The results demonstrated that the edible leafy spinach plant can uptake, accumulate, and metabolize paracetamol, which is one of the most commonly used over-the-counter drugs. Despite the bioremediation process (in planta and microbial), higher doses (100 mg/L and 200 mg/L) of paracetamol and its formed metabolites induced phytotoxicity, causing oxidative stress and irreversible damage to spinach roots and edible shoots. Accumulation of these pseudo-persistent pharmaceutical pollutants in the water, soil, and plant matrix over time may increase the population of emerging opportunistic pathogens in biofilms, have a high risk for horizontal gene transfer, generate new multiple-drug-resistant strains, and thus pose a direct threat to animal and human health via the food chain. 

These findings could facilitate the development of guidelines for improving wastewater treatment and utilization methods. Additionally, our research also provides a new perspective exploring the synergistic roles of plants and associated beneficial microbes in promoting plant growth and enhancing the bioremediation process of contaminants of emerging concern (CEC) in the environment. The fast growth of spinach and its fast uptake and degradation of paracetamol recommend it as a potential plant for the phytoremediation of polluted lands. The harvested plants could be used as a source of biofuel.

Our extended research plan will offer a detailed evaluation and deeper understanding of the biochemical and molecular repercussions of these pollutants and contribute to addressing the fears arising in the flora and microbial community for endorsing the application of treated wastewater to promote sustainable agricultural development in the United Arab Emirates.

## Figures and Tables

**Figure 1 plants-11-01626-f001:**
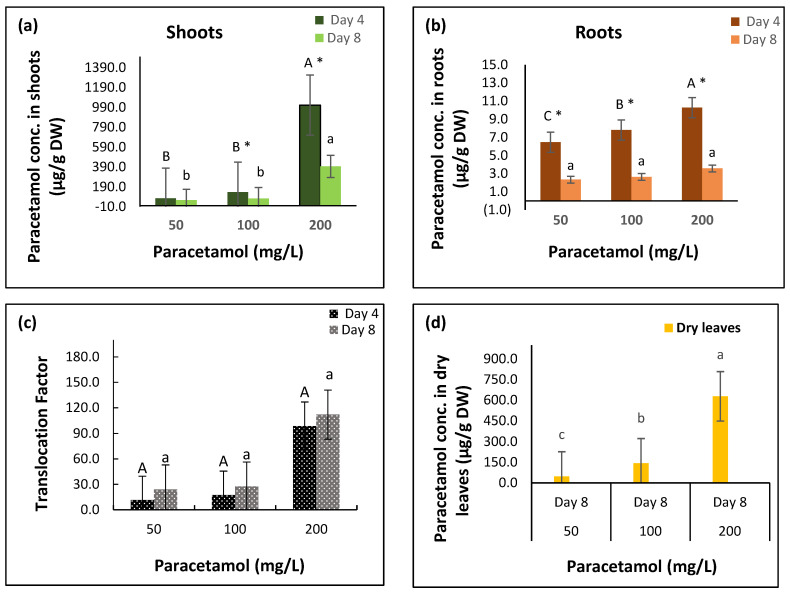
Effect of different paracetamol treatments at different time points on the concentration of paracetamol (µg/g DW) in spinach (**a**) shoots, (**b**) roots, (**c**) translocation factor, and (**d**) dry leaves. Error bars represent means ± S.E. of three biological replicates. Means with different upper-case and lower-case letters indicate significant differences (*p* ≤ 0.05) between the different paracetamol treatments at four and eight days, respectively. Asterisks (*) indicates significant differences between four and eight days at a certain concentration.

**Figure 2 plants-11-01626-f002:**
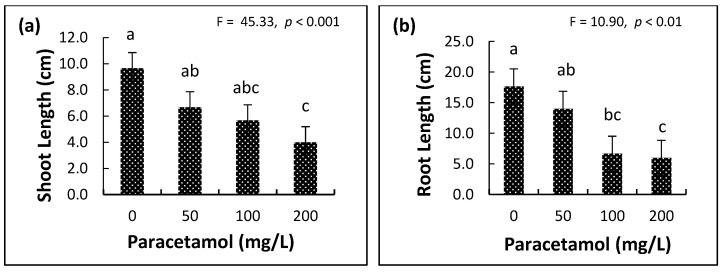
Effects of different paracetamol treatments on spinach (**a**) shoot length, (**b**) root length and (**c**) number of leaves, and (**d**) growth tolerance index (%) of spinach shoots (GTIS) and roots (GTIR) after eight days. Error bars represent means ± S.E. of three biological replicates. For subfigures (**a**–**c**), means with the different letters are significantly different at *p* ≤ 0.05. For subfigure (**d**), different upper-case and similar lower-case letters indicate significant differences between GTIS and non-significant differences between GTIR, respectively, at the different paracetamol treatments.

**Figure 3 plants-11-01626-f003:**
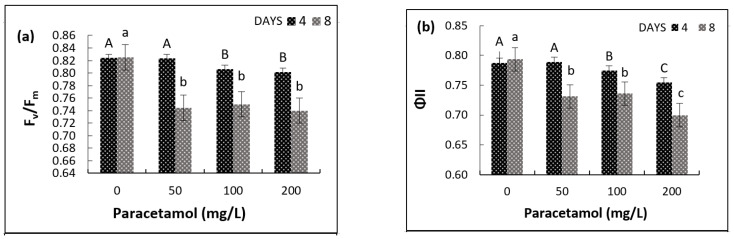
Effects of different paracetamol treatments and exposure periods on (**a**) F_v_/F_m_, (**b**) quantum yield of electron transport in photosystem II (ΦII), (**c**) Chl a, (**d**) Chl b, (**e**) carotenoids and (**f**) total chlorophyll (mg/g FW). Error bars represent ± S.E. of three biological replicates. Means with different upper-case and lower-case letters indicate significant differences (*p* ≤ 0.05) at days four and eight, respectively.

**Figure 4 plants-11-01626-f004:**
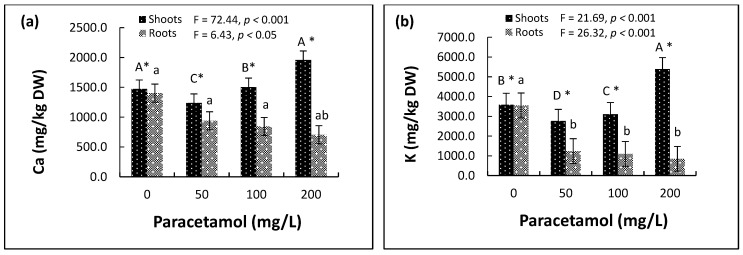
Effects of different paracetamol treatments on concentrations (mg/kg DW) of (**a**–**c**) macronutrients, (**d**,**e**) micronutrients and (**f**) sodium in shoot and root systems of spinach. Error bars represent means ± S.E. of three biological replicates. Means with different upper-case and lower-case letters indicate significant differences (*p* ≤ 0.05) in nutrient and sodium concentrations between shoots and roots, respectively, at the different paracetamol treatments. Asterisks (*) indicate significant differences (*p* ≤ 0.05) in the concentration of a nutrient or sodium between shoots and roots at a certain paracetamol level.

**Figure 5 plants-11-01626-f005:**
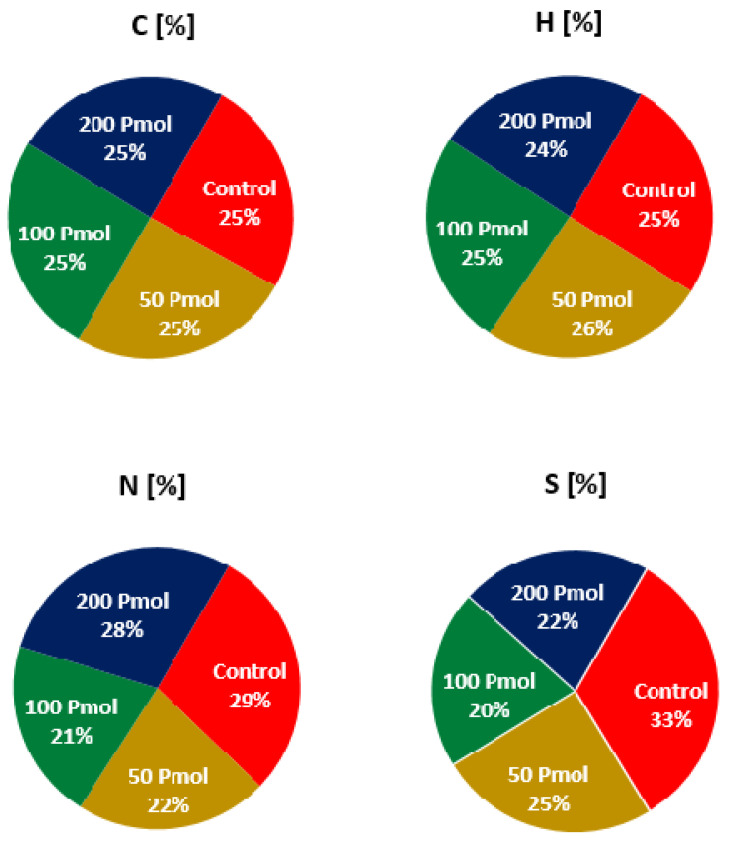
Variation in weight percentages of C, H, N, and S in spinach shoots after eight days of paracetamol treatment.

**Figure 6 plants-11-01626-f006:**
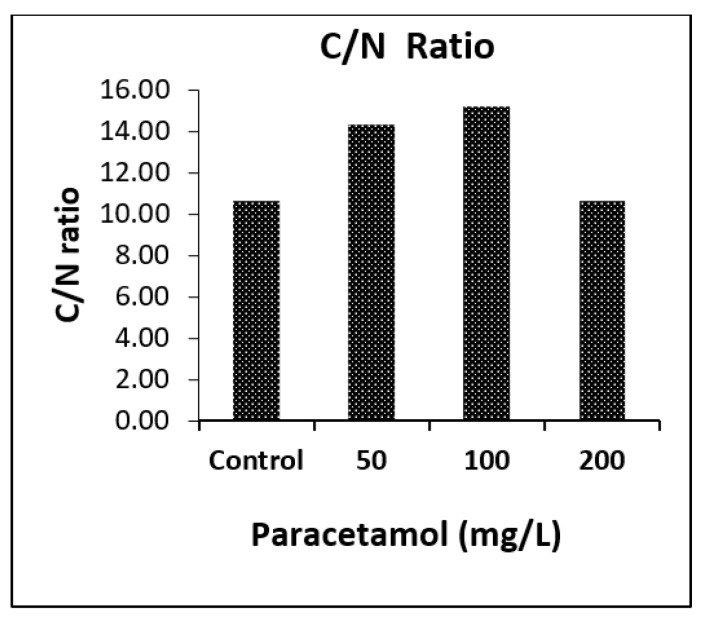
Variation in C/N ratio in spinach shoots after eight days of paracetamol treatments.

**Table 1 plants-11-01626-t001:** Results of two-way ANOVAs showing the effects of different paracetamol treatments, exposure periods, and their interactions on the concentration of paracetamol (µg/g DW) in spinach shoots, roots, and translocation factor.

Variable	Factor	df	F-Ratio	*p*-Value
Shoots	Period (P)	1	488.82	<0.001
	Treatments (T)	2	1538.01	<0.001
	P × T	2	332.65	<0.001
Roots	Period (P)	1	587.76	<0.001
	Treatments (T)	2	45.42	<0.001
	P × T	2	11.49	<0.01
Translocation Factor	Period (P)	1	26.52	<0.001
	Treatments (T)	2	591.35	<0.001
	P × T	2	0.17	ns

ns: non-significant at *p* ≤ 0.05.

**Table 2 plants-11-01626-t002:** Results of two-way ANOVAs showing the effects of different paracetamol treatments, exposure periods, and their interactions on chlorophyll fluorescence (F_v_/F_m_ and ΦII), Chl a, Chl b, carotenoids, and total chlorophyll content (mg/g FW) in spinach.

Variable	Factor	df	F-Ratio	*p*-Value
F_v_/F_m_	Period (P)	1	29.45	<0.001
	Treatment (T)	3	7.07	<0.01
	P × T	3	3.66	<0.05
ΦII	Period (P)	1	21.07	<0.01
	Treatment (T)	3	10.93	<0.01
	P × T	3	3.43	<0.05
Chl a	Period (P)	1	6.19	<0.05
	Treatment (T)	3	1.77	ns
	P × T	3	1.52	ns
Chl b	Period (P)	1	10.02	<0.01
	Treatment (T)	3	2.32	ns
	P × T	3	2.31	ns
Carotenoids	Period (P)	1	4.34	ns
	Treatment (T)	3	1.70	ns
	P × T	3	1.68	ns
Total Chlorophyll	Period (P)	1	7.11	<0.05
	Treatment (T)	3	1.89	ns
	P × T	3	1.71	ns

ns: non-significant at *p* ≤ 0.05.

**Table 3 plants-11-01626-t003:** Microbial consortium of spinach roots treated with paracetamol for eight days.

Treatment	Isolate and Colony Description	Organism	Probability (%)	Confidence	Role in Plants
Control spinach roots	Gram +ve micrococcus, yellowish moist, moderate size	*Kocuria kristinae*	98	Excellent Identification	Antagonist bacteria in spinach [32].
	Gram −ve, white, large flat, and dry	*Pasteurella pneumotropica*	90	Good Identification	Endophytic bacterium capable of fixing nitrogen and solubilizing phosphate [33].
	Gram −ve, moderate moist	*Comamonas testosteroni*	99	Excellent Identification	Spinach microbiota [34]. Involved in the degradation of aromatic compounds [35].
	Gram −ve, small moist	*Acinetobacter Iwoffii*	99	Excellent Identification	Spinach microbiota [34,36]. Involved in the degradation of paracetamol and aromatic compounds [35].
50 mg/L paracetamol-treated spinach roots	Gram −ve, yellow, large, and dry	*Burkhulderia cepacia* group	95	Very Good Identification	Spinach microbiota [36]. Increased growth parameters in *Zea mays* [37].
	Gram −ve, pale yellow, small, moist	*Pseudomonas florescens*	98	Excellent Identification	Involved in the degradation of paracetamol and aromatic compounds [35], Spinach microbiota [36]. Phosphorus-solubilizing, protease production in *Phragimates australis* [38]. Salt tolerant in groundnut (*Arachishypogaea*) [39]. Antifungal, plant growth promotion. Increased growth parameters, increased Pb uptake, root elongation in *Brassica napus* and *Solanum nigrum* [40].
	Gram +ve, white, large and moist	*Staphylococcus* *haemolyticus*	95	Very Good Identification	Spinach microbiota [36]. Increased growth parameters in *Triticum aestivum* [41].
	Gram −ve rods, pale yellow	*Stenotrophomonas* *maltophilia*	95	Very Good Identification	Spinach microbiota [36]. Increased growth parameters in *Triticum aestivum* [41].
100 mg/L paracetamol-treated spinach roots	Gram +ve, pale yellow, moderate, moist	*Kocuria kristinae*	87	Acceptable Identification	Antagonist bacteria in spinach [32].
	Gram +ve, white, small, round	*Kocuria rosea*	98	Excellent Identification	Capable of growing on naphthalene, phenanthrene and fluoranthene, on all three polycyclic aromatic hydrocarbons (PAHs) [42].
200 mg/L paracetamol-treated spinach roots	Gram −ve, coccobacillus, yellow, large and dry colonies	*Brucella melitensis-* Highly pathogenic	91	Good Identification	Foodborne pathogen. Contaminant of leafy vegetables and causes human brucellosis [43,44].
	Gram −ve bacilli, yellow, large and moist colonies	*Sphingomonas paucimobilis*	92	Good Identification	Spinach microbiota [35]. Increase in growth parameters in *Triticum aestivum* [41]. Involved in the degradation of paracetamol and aromatic compounds [35]. Spinach microbiota, involved in PPCP biodegradation, produces lignin-degrading enzymes [36,45].
	Gram +ve, white, moderate size	*Staphylococcus auricularis*	98	Excellent Identification	Spinach microbiota, increases growth parameters in *Triticum aestivum* [36,41].

**Table 4 plants-11-01626-t004:** Microbial consortium of spinach shoots treated with paracetamol for eight days.

Treatment	Isolate and Colony Description	Organism	Probability (%)	Confidence	Role in Plants
Control spinach shoots	Gram +ve, white moist, moderate size colonies	*Staphylococcus hominis ssp hominis*	N.A.	Low Discrimination Organism	Epiphytic bacteria from fruits and leafy greens, spinach microbiota [36], potential biocontrol agents, able to reduce the proliferation of *E. coli* O157:H7 and *S. enterica* in fruits and vegetables [46].
	Gram +ve, white moist, moderate size colonies	*Aerococcus viridans*	N.A.	Low Discrimination Organism	Epiphytic bacteria on leafy greens, capable of fixing nitrogen and solubilizing phosphate [33].
	Gram −ve, moderate yellow moist colonies	*Oligella ureolytica*	N.A.	Low Discrimination Organism	Pathogenic bacteria [47].
	Gram −ve, moderate yellow moist colonies	*Aeromonas salmonicida*	NA	Low Discrimination Organism	Plant growth promoting rhizobacteria, involved in biodegradation of xenobiotic compounds from contaminated water/soil environment [38,48].
50 mg/L paracetamol-treated spinach shoots	Gram −ve bacilli, yellow, large and moist colonies	*Sphingomonas paucimobilis*	89	Good Identification	Spinach microbiota, involved in the degradation of paracetamol and aromatic compounds [34,35,36].
	Gram +ve, white, moderate size	*Staphylococcus hominis ssp hominis*	95	Very Good Identification	Spinach microbiota [36]. Epiphytic bacteria from fruits and leafy greens are potential biocontrol agents, able to reduce the proliferation of *E. coli* O157:H7 and *S. enterica* in fruits and vegetables [46].
100 mg/L paracetamol-treated spinach shoots	Gram +ve, pale yellow, moderate, moist	*Kocuria rosea*	98	Excellent Identification	Capable of growing on naphthalene, phenanthrene and fluoranthene, on all three polycyclic aromatic hydrocarbons (PAHs) [42].
	Gram −ve, white, moderate size, moist	*Escherichia coli*	93	Very Good Identification	Foodborne pathogen, spinach microbiota [34].
200 mg/L paracetamol treated spinach shoots	Gram +ve, white, moderate, moist	*Staphylococcus vitulinus*	98	Excellent Identification	Spinach microbiota [34,36].
	Gram +ve, white, moderate, moist	*Gamella bergeri*	90	Good Identification	Epiphytic bacteria from fruits and leafy greens are potential biocontrol agents, able to reduce the proliferation of *E. coli* 157:H7 and *S. enterica* in fruits and vegetables [46].

## Data Availability

The data are available within the article and Appendix A.

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
