# Peer review of "Assessment of Uptake, Accumulation and Degradation of Paracetamol in Spinach (Spinacia oleracea L.) under Controlled Laboratory Conditions"

_plants, 2022, doi:10.3390/plants11131626_

Round 1

Reviewer 1 Report

Review Remarks on MS: entitled: ‘Assessment of Uptake, Accumulation and Degradation of Paracetamol in Spinach (Spinacia oleracea L.) under Controlled Laboratory Conditions.’

The following are some general points and scientific queries that need to be addressed in the manuscript:

·         Please follow the standard guidelines for the presentation of scientific names of plants and microorganisms throughout the manuscript.

·         Please concise the introduction and remove general information.

·         What is the purpose of Table 8 in the manuscript because the authors have already provided enough literature? Please pay attention.

·         Please check the numbering of tables in the manuscript.

·         Please check line 86 and rephrase it. Also, check the manuscript for minor errors critically.

·         The ANOVA summary can be placed with respective Figures. Please shift some figures and tables to the supplementary part to avoid repetition.

·         Please provide the values of LOD and LOQ.

·         Please concise the materials and methods part.

Author Response

Dear Reviewer,

Thank you for your valuable comments to improve the manuscript.

Please see the attachment for responses. 

Regards,

Zarreen Badar

Reviewer 2 Report

This manuscript presents the impact of paracetamol (acetaminophen) on selected growth, biochemical, and phytochemical parameters of spinach grown in hydroponic conditions. The subject of this paper is well within the scope of the journal. The data presented will add interesting information about the effects of this pharmaceutical substance on the growth, some physiological parameters, and paracetamol accumulation and translocation in plants. However, there are several issues that need to be corrected before the manuscript will be accepted for publication.

Specific comments:

11) Latin names should always be italicized.

22) Introduction, lines 128-134: English plant species names should not be capitalized.

33) The introduction is clearly and exhaustively written. However, there is no research hypothesis. The research hypothesis should take the form of a statement (not a question or guess). The hypothesis should always explain what you expect to happen. Rejecting the null hypothesis and accepting the alternative hypothesis is the basis for building a good research study.

44) In the figures (y-axis descriptions) and in tables it should be clearly indicated whether the content of substance or element is given in terms  of fresh or dry plant weight.

55) Figure 2: Shoot and root lengths or number of leaves were only measured in 3 plants (n=3; 3 biological replicates)?  Such results are unbelievable, did you not have any more plants to measure?

66) Figure 3 e: instead of PS II on y axis should be ‘(ΦII)’

7) In the results the authors wrote: ‘In the roots, there were gradual decreases in the concentrations of the three macronutrients (Ca, Na and K)…’ but sodium has never been classified as essential element from the group of macronutrients in plants! Please, correct it here and further in the text.

8) In the discussion, the paracecamol concentration used in the study should be compared to the concentration that may be present in the environment contaminated with this substance. On what basis and why did you choose these concentrations for the experiment?

99) Reference in the discussion to bisphenol a is unnecessary because paracetamol and BPA are chemically different substances.

110)  M&M: which cultivar of Spinacia oleracea was used? Why the nutrient solution had only 20% concentration? At what temperature, photoperiod, PAR intensity the plants grew in hydroponic conditions?  

111)  Please, check the reference list for proper formatting (are you sure all these items are needed? there are quite a lot of them).

Author Response

(The authors gave the same response as above.)

Round 2

Reviewer 1 Report

After careful review, I found the authors have addressed all the comments and they are satisfactory.

Author Response

Dear Reviewer,

Thank you very much for your positive feedback.

Best Regards,

Zarreen Badar

Reviewer 2 Report

Dear Authors,

The statement that sodium is a micronutrient is also not correct. Sodium is not included in the group of essential elements (neither macro- nor micronutrients). It was only considered a beneficial element (for some plant species). This has to be revised again as it is a serious error in the classification of essential elements for plants.

Author Response

Dear Reviewer,

Thank you very much for your insightful suggestions and positive feedback.

Best Regards,

Zarreen Badar
